# Navigating public, semi-public, and private drug use environments: A qualitative examination of unsheltered homelessness and drug use in Baltimore City

Ashley Q. Truong[1]*, Bridget Duffy[2], Haneefa T. Saleem[3], Jessie Chien[4], Gregory D. Kirk[5], Shruti H. Mehta[5], Becky L. Genberg[5], Sabriya L. Linton[6]

**1** Department of Population Health, NYU Grossman School of Medicine, New York, New York, United States of America, **2** Center for Injury Research and Policy, The Abigail Wexner Research Institute at Nationwide Children's Hospital, Columbus, Ohio, United States of America, **3** Department of International Health, Johns Hopkins Bloomberg School of Public Health, Baltimore, Maryland, United States of America, **4** Department of Community Health Sciences, UCLA Fielding School of Public Health, Los Angeles, California, United States of America, **5** Department of Epidemiology, Johns Hopkins Bloomberg School of Public Health, Baltimore, Maryland, United States of America, **6** Department of Mental Health, Johns Hopkins Bloomberg School of Public Health, Baltimore, Maryland, United States of America

* ashley.truong@nyulangone.org

## Abstract

Unsheltered people experiencing homelessness (PEH) in the United States (US) have a disproportionately high burden of illegal drug use and overdose. Due to their economic and social marginalization, unsheltered PEH who use drugs are often constrained in where they use drugs, and this can hinder their engagement in harm reduction practices and increase their overdose risk. Using interpretative phenomenological analysis, this study investigates how unsheltered PEH who use drugs in Baltimore City perceive and engage with their drug use environment and how public, semi-public, and private features of their environment influence drug use practices and overdose risk. Public settings (e.g., street) are accessible to the general public, while semi-public settings (e.g., library restrooms) provide public access with some restrictions. In contrast, private settings (e.g., private residence) are under private ownership with no public access. Participants were recruited from the AIDS Linked to Intravenous Experience (ALIVE) Study. Data was analyzed from interviews conducted with nine participants. Participants described complex experiences managing their drug use in public, semi-public, and private spaces. In public and semi-public spaces, participants engaged in varied strategies to seek privacy. Privacy in these spaces provided participants a sense of refuge from policing and interference, but in some instances, at the expense of drug use safety. To access private spaces for drug use and shelter, participants reported engaging in an informal economy of sharing drugs and other resources. Using in private spaces enhanced participants' sense of comfort and enabled safer drug use practices, though access to these spaces

**Data availability statement:** There are ethical restrictions on sharing de-identified data from this study because the data contains potentially identifying and sensitive patient information. These restrictions are imposed by the Johns Hopkins Institutional Review Board. De-identified data requests may be sent to the ALIVE study PIs, Shruti Mehta (smehta@jhu.edu) and Gregory Kirk (gdk@jhu.edu), or the Johns Hopkins Bloomberg School of Public Health Institutional Review Board (BSPH.irboffice@jhu.edu).

**Funding:** This study was funded by the National Institute on Drug Abuse [U01DA036297, PIs: Mehta/Kirk; R01DA053136, PI: Genberg], the Johns Hopkins Bloomberg School of Public Health Center for Qualitative Studies in Health and Medicine's Dissertation Enhancement Award, and The Paul V. Lemkau Scholarship Fund. AQT was supported by the National Institute on Drug Abuse [T32DA007292-31A1; PIs: Johnson/Maher].

**Competing interests:** The authors have declared that no competing interests exist.

was inconsistent. A building where people can use drugs safely and privately with supervision from peers and medical staff was commonly described as an ideal drug use space across participants. Findings highlight the need to identify resources and interventions to facilitate safer drug use for unsheltered PEH who use drugs in the US. Future interventions should take a multi-level, harm reduction approach, targeting contextual and individual factors to promote safer drug use and minimize the risk of drug use-related harms among unsheltered PEH.

## Introduction

The affordable housing crisis in the United States (US) is a critical public health issue [1]. Rising housing costs and long public housing waitlists have contributed to the growing number of people who are unstably housed, and a recent US Supreme Court case ruling, Grants Pass v. Johnson, now allows jurisdictions to issue fines and arrest people for sleeping or camping in public spaces [2,3]. Concurrently, the US is in the midst of an overdose crisis, with nearly 110,000 overdose deaths documented in 2023 [4]. Evidence indicates that people experiencing homelessness (PEH) have a higher burden of illegal drug use and overdose compared to the general population [5–8].

The relationship between drug use and homelessness is complex and involves a combination of individual, interpersonal, and structural factors [9]. Drug use and homelessness share many determinants, such as adverse childhood experiences and poor mental health [10–14]. Moreover, homelessness may precipitate drug use and exacerbate current use [10,15], as PEH have reported using drugs to regulate their body temperature, treat physical pain stemming from their lack of housing, and cope with emotional pain related to experiencing homelessness [15–17]. Conversely, drug use may increase housing instability through straining relationships with those who provide informal housing support and through discriminatory housing policies that preclude people who use drugs (PWUD) from accessing housing assistance and other housing supports [10,18,19]. At the structural level, PEH who use drugs are subject to the dual stigmatization and criminalization of homelessness and substance use disorders [16,20–24], which may increase their interactions with the criminal legal system and experiences of violence, hinder engagement in healthcare services, and promote drug use behaviors (e.g., rushing injections) that increase vulnerability to overdose [25–28].

Experiences of homelessness can be further categorized as experiences of sheltered homelessness and of unsheltered homelessness [2]. Based on the US Department of Housing and Urban Development's (HUD) definitions, people experiencing sheltered homelessness are those who sleep in emergency shelters, transitional housing, or supportive housing while unsheltered PEH reside in places not typically used as regular accommodations such as cars, parks, encampments, and vacant buildings [2]. Due to increased exposure to weather, violence, and social and economic marginalization, unsheltered PEH have a higher mortality rate and poorer physical, mental, and sexual health compared to those who are experiencing sheltered homelessness [29].

To evaluate the role of environmental conditions in influencing drug use behaviors and related health harms, there has been a proliferation of literature guided by Rhodes' risk environment framework investigating the "micro-injecting environments" of people who inject drugs (PWID), with a particular focus on drug use in public and semi-public settings [25,30–37]. According to Rhodes' risk environment framework, individual drug use behaviors are shaped by a dynamic interplay of micro-level (e.g., individual and interpersonal) and macro-level (e.g., social structures and systems) factors constituting the 'risk environment' [38]. Quantitative studies of PWUD's micro-level environments have found that injecting drugs in public and semi-public settings is associated with rushed drug use, use of unfiltered water, receptive syringe sharing, and sequelae of injection drug use, including non-fatal overdose and HIV acquisition [25,35,37,39–41]. Qualitative studies have identified fear of police interaction as a primary driver of rushed injection drug use in public and semi-public settings [41,42].

Although prior studies have consistently demonstrated that experiencing homelessness is significantly associated with public injection drug use [31,34,35], few studies have qualitatively explored how micro drug use environments shape drug use behaviors among PEH and none have done so in the US [43]. Furthermore, despite research documenting that risk factors (e.g., police interactions) in micro-level drug use environments may be exacerbated for unsheltered PEH who only have access to visible and public and semi-public settings compared to sheltered PEH and the general population, there is a dearth of studies qualitatively examining the intersecting phenomena of unsheltered homelessness and drug use [17,44]. These existing studies are limited to European settings [17,44]. They likely do not reflect changing temporal trends related to the rise of fentanyl and differing sociopolitical contexts of drug use and unsheltered homelessness in North America [45–48].

The present study addresses these gaps by investigating micro drug use environments among unsheltered PEH in Baltimore City, including how they perceive and engage with their environment and how public, semi-public, and private spaces shape drug use practices and overdose risk.

## Methods

We use interpretative phenomenological analysis (IPA) to explore the intersection of unsheltered homelessness and drug use, producing an in-depth analysis of participants' lived experiences.

### Study setting

This study was conducted in Baltimore City, a mid-Atlantic post-industrial city. Among jurisdictions in the US with at least 500,000 people and among all jurisdictions in Maryland, Baltimore City has the highest rate of fatal overdose [49,50]. According to the 2023 Baltimore City Point-in-Time (PIT) Count, approximately 92% of the 1,550 PEH reported experiencing sheltered homelessness (including residing in emergency shelters and transitional housing), and less than 7% were unsheltered [51]. The most commonly reported sleeping places among unsheltered PEH surveyed as part of the PIT Count were the street or sidewalk, abandoned building, and woods or outdoor encampment [51].

### Participants and procedures

According to Smith et al., sample sizes in studies using IPA are necessarily small due to the approach's focus on understanding the particularities of participants' daily lived experiences, and to enable deep analysis of each interview [52]. As such, larger sample sizes are typically discouraged in IPA studies, and recommended sample sizes range from 4–10 [53,54]. A total of nine individuals were included in the study. Although we conducted interviews with 12 participants, three were excluded from the analysis because they were ended early or due to concerns about the participants' mental health during the interview. Specifically, two interviews were ended early because one participant was too lethargic to continue and one participant was losing her voice and what she said could not be clearly heard. A third interview was excluded because the participant expressed paranoia and what he discussed was consequently difficult to understand. Interviews lasted about an hour on average.

We employed purposive homogenous sampling to obtain a fairly homogenous sample of participants in terms of unsheltered housing status and recent illegal drug use [55]. Eligibility criteria for the present study included 1) using illegal drugs (e.g., cocaine alone, heroin alone, "speedball" which refers to use of cocaine and heroin together, non-prescribed/illicitly manufactured fentanyl, crystal methamphetamine, xylazine, hallucinogens, or non-prescribed prescription medication) in the past 30 days and 2) sleeping in an unsheltered situation (e.g., street or sidewalk; abandoned building; park; car; encampment; tent; under a bridge, highway, or tunnel; bus or train station) for at least 30 days in the past 6 months.

Participants were recruited from the ongoing AIDS Linked to Intravenous Experience (ALIVE) study, a prospective community-based cohort study of people with a history of injection drug use in Baltimore City [56]. First established in 1988, the ALIVE cohort has undergone five additional enrollment periods. Eligibility criteria for the ALIVE study included being at least 18 years of age and self-reporting a history of injection drug use. From October 2023 through March 2024, ALIVE staff referred interested ALIVE participants when they came in for their first or semi-annual study visit. Referred ALIVE participants then completed a brief screener with AQT in-person at the time of their study visit or over the phone following their study visit to determine their eligibility.

AQT conducted one-on-one, in-person interviews with participants. Oral informed consent was obtained from all participants prior to beginning the interview. A semi-structured in-depth interview guide informed by the research question and risk environment framework was developed to explore participants' life history and social identity; experiences with unsheltered living; history of substance use; how they interpret their drug use spaces; and policy recommendations for housing and safe consumption spaces [38]. Screening, consenting, and data collection occurred in a private room at the ALIVE study clinic. Participants were compensated with a $40 check after completing their interview.

## Data analysis

Participant interviews were audio recorded and transcribed verbatim, and transcripts were reviewed for accuracy. AQT and BD used IPA to analyze data, following seven steps outlined by Smith et al. (Fig 1): 1) reading and re-reading the transcript; 2) exploratory noting of the transcript, including key phrases participant used or AQT and BD's conceptual thoughts; 3) consolidating exploratory notes into experiential statements; 4) searching for connections across experiential statements and organizing them into clusters; 5) creating personal experiential themes (PETs) to identify each cluster, 6)

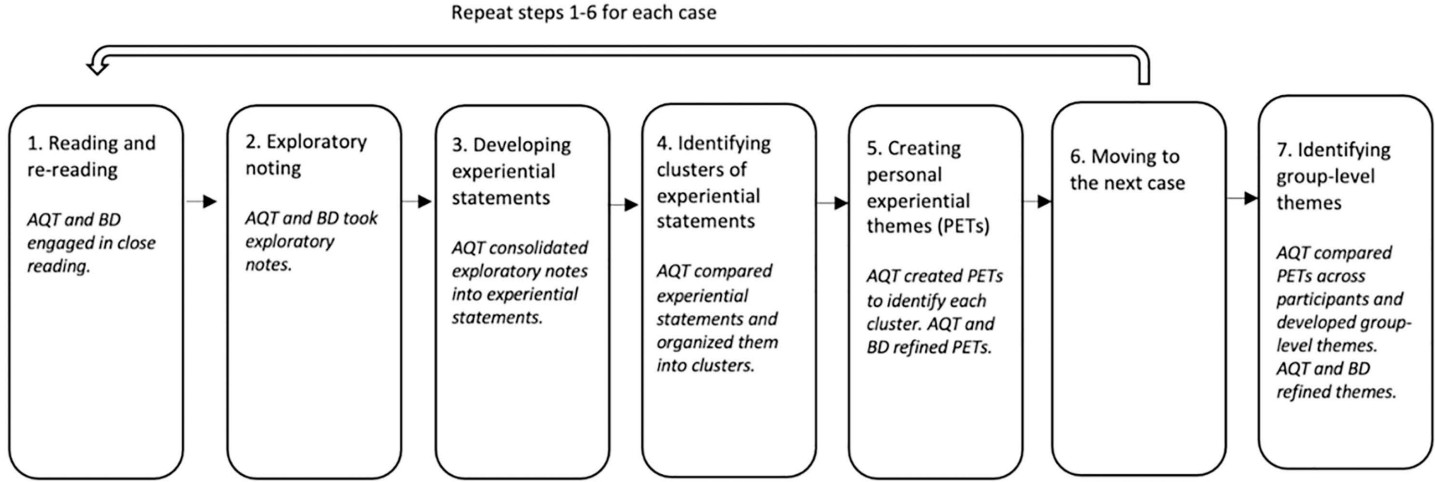

**Fig 1. Interpretative phenomenological analysis steps.**

moving to the next case, and 7) identifying group-level themes based on similarities across PETs and the research question [52].

### Ethical Approval

The study was approved by the Johns Hopkins Bloomberg School of Public Health Institutional Review Board.

## Results

The analytic sample consisted of six male participants and three female participants. In total, five participants identified as non-Hispanic Black, three participants identified as non-Hispanic white, and one participant identified as American Indian. The average age of the sample was 48 years. To help contextualize participants' experiences, sociodemographic information, the types of drugs used, and unsheltered sleeping places are summarized in Table 1. To protect participants' identities, we anonymized participants' names.

Participants described complex experiences managing their drug use in public, semi-public, and private spaces. Across all cases, we identified three group-level themes from the data: 1) "Privacy in public and semi-public spaces"; 2) "Experience accessing private spaces"; and 3) "Desired drug use spaces: overdose prevention sites".

### Theme 1: Privacy in public and semi-public drug use spaces

Consistent across drug use spaces—public, semi-public, and private—participants intonated a desire for privacy when using drugs. However, not all participants had access to private spaces. Consequently, all participants described using drugs in public and semi-public settings. Commonly reported public and semi-public drug use spaces include the street, alleys, public transit, vacant buildings, and library or restaurant bathrooms.

Privacy was not guaranteed in public and semi-public places, as participants had little control over their visibility and the social and physical dimensions of such places. As a result, participants described employing various methods to establish personal space where they could conceal their drug use in these settings. Although participants sought out hidden places to use drugs in public and semi-public settings, they were also aware of the dangers inherent to using in these hidden spaces, including overdose and violence.

**Subtheme 1.1: Seeking privacy in public and semi-public spaces.** Participants portrayed seeking privacy in public and semi-public spaces as a precarious process. Despite establishing personal space that increased their sense of privacy in public and semi-public drug use settings, participants remained acutely aware of their visibility and the risk of being seen by police, neighborhood residents, and other people occupying the space. For example, several participants described positioning themselves in the back of buses and train cars to establish personal space and enhance their sense of privacy when using drugs. Gary recounted:

> I would pick a car on the train, like a train car, where there weren't many people or anybody even near the area where I was at. And I would just lean down in the seats in the back and kind of put myself in the corner and do it right there. And then nobody would see me [...] I would decide to do it [use drugs] because I would survey the area and see if it was safe. Am I able to do it without having any interference from somebody trying to not just rob me but getting in the way or bugging me for drugs or if I can have a spot that I can have peace of mind as I do it.

Gary captures his dynamic thinking process when using on trains. At the crux of his thinking process is a keen sense of spatial awareness, demonstrated by his description of how he positions himself on the train to avoid being seen by others. To further conceal his drug use from the public, he uses his body and the physical walls of the train car to create a barrier between his drug use and the outside world. Like several other participants, Gary also described taking extra caution to conceal his drug use from police:

**Table 1. Demographic, housing, and drug use characteristics of participants.**

| Pseudonym | Sex | Race | Housing[a] | Drug use[b] |
|---|---|---|---|---|
| Edith | Female | Black | • Alley<br>• Encampment<br>• Park<br>• Street or sidewalk | • Cocaine<br>• Fentanyl<br>• Heroin<br>• Non-prescribed methadone |
| Eric | Male | White and other | • Tent | • Cocaine<br>• Heroin |
| Frank | Male | Black | • Abandoned building<br>• Car<br>• Park | • Cocaine<br>• Fentanyl<br>• Heroin<br>• Non-prescribed sedative<br>• Non-prescribed tranquilizer<br>• Xylazine |
| Gary | Male | White | • Abandoned building<br>• Bus or train station<br>• Street or sidewalk | • Cocaine<br>• Crystal methamphetamine<br>• Fentanyl<br>• Hallucinogen<br>• Heroin<br>• Speedball<br>• Xylazine |
| Jeff | Male | Black | • Bus or train station<br>• Car<br>• Park<br>• Tent | • Crystal methamphetamine |
| Lisa | Female | White | • Abandoned building<br>• Car<br>• Park<br>• Train | • Cocaine<br>• Heroin<br>• Non-prescribed buprenorphine |
| Mark | Male | White | • Park | • Cocaine<br>• Fentanyl |
| Robert | Male | Black | • Bus or train station<br>• Abandoned building<br>• Car<br>• Street or sidewalk<br>• Under a bridge, highway, or tunnel | • Cocaine<br>• Fentanyl<br>• Heron<br>• Non-prescribed buprenorphine<br>• Speedball |
| Tanya | Female | American Indian | • Abandoned building<br>• Car<br>• Street or sidewalk | • Crystal methamphetamine<br>• Cocaine<br>• Fentanyl<br>• Heroin<br>• Speedball |

[a]Unsheltered sleeping situations in the past 6 months captured with the question: Have you slept in any of the following places in the past 6 months for a total of at least 30 nights? Response options included: street or sidewalk; abandoned building; park, car; encampment, tent city, or homeless camp; tent outside; under a bridge, highway, or tunnel; bus or train; and other (specify).

[b]Drugs used in the past 30 days captured with the question: Have you used any of the following drugs in the past 30 days? Response options included: cocaine; heroin; speedball; fentanyl, crystal methamphetamine; xylazine; hallucinogens; methadone that wasn't prescribed to you; buprenorphine that wasn't prescribed to you; prescription painkillers that weren't prescribed to you; prescription sedatives that weren't prescribed to you; prescription tranquilizers or antianxiety drugs that weren't prescribed to you; and other (specify).

I'm very careful about how I do it [use drugs]. I make sure right before I'm about to get high that whatever I use to make it so I can get high is put away, and also I make sure I'm not visible or in a area where they might smell the smoke from my pipe or see me smoking it. Also, if they [police] are coming, I have a way to get out of there before they catch me because if they're coming in there and arrest one person, they're going to take everybody. So it's constantly dealing with that.

Gary's cognitive process of safety planning included concealing signs of his drug use that may alert police and developing an escape route should police enter the place he was using drugs. The desired level of privacy from police varied based on context and how participants presented—as a PWUD or unsheltered PEH. Similar to Gary, many participants cited fear of surveillance and police interaction as reasons for seeking privacy in public and semi-public drug use spaces. On the other hand, in the context of their unsheltered homelessness, many participants reported neutral to positive perceptions of and experiences with police. Indeed, some participants preferred to shelter near police or security personnel, noting that police and security presence enhanced their sense of safety when sleeping on the streets. The perceived differential and harsher treatment by police depending on whether a participant presented as a PWUD or PEH in public and semi-public settings is well summarized in Mark's interview when he explained why he prefers to not use drugs in the park where he sleeps:

> You don't shit where you eat. I don't use where I sleep [...] Because if it—that cause, that cause problems because if somebody sees me using where I'm sleeping then they might call the police. Then they'll start messing with me. I'll be the guy, the junkie now. Not the homeless guy, so I don't want to do that.

This passage underscores the intersectionality of Mark's identities as a PWUD and an unsheltered PEH. Using drugs where he sleeps makes his identity as a PWUD more prominent to those in the vicinity and transforms people's perception of him as "the homeless guy" into "the junkie." Consistent with other participant interviews, Mark highlights the role of community surveillance in precipitating police interactions.

Notably, a couple participants described inequitable policing across neighborhoods that make finding privacy challenging when in different public and semi-public spaces. Robert, for example, provided a hypothetical example to illustrate targeted policing activity on the Johns Hopkins medical campus and its impact on his treatment in nearby public spaces:

> Well, this area right here [adjacent to Johns Hopkins' medical campus], the city cops would definitely stop you from just standing out in front of a building or a business or something like that. Even though the people inside of the business aren't even worried about it, he would make it his business.

Robert goes on to attribute the inequitable policing to community development in East Baltimore, linking markers of gentrification, Johns Hopkins, and more punitive policing practices on the medical campus:

> John Hopkins is expanding, especially in this area. If you walk up certain streets, you have the security in booths and stuff like that. Like I said, you have students around here, and then you have the luxury apartments and stuff like that. So they they more so want to keep Johns Hopkins area a little bit cleaner because it's one of the bigger, more prominent places in East Baltimore [...] And you don't want to get caught around the hospital or something using or selling. It's like selling drugs in a school zone.

**Subtheme 1.2: Risks and benefits of privacy in public and semi-public spaces.** Participants had nuanced perceptions of the risks and benefits of privacy in public and semi-public drug use settings. Using drugs in obscured public and semi-public spaces was perceived positively, helping to prevent unwanted interference from police, PWUD, and others in the vicinity. In particular, fear of police interaction or arrest led some participants to rush their drug use in public and semi-public spaces. For example, Tanya describes her motivations for rushing injections and suggests that the context of her drug use and consequences of policing differ by jurisdiction (i.e., Baltimore County versus Baltimore City):

> Interviewer: And do you ever feel like you have to rush your injections?
>
> Tanya: Oh, yeah. Especially when you're outside because you don't want, like I said, anybody to see you or call the police or anything like that. So you have to rush. You got to hurry it along because you'll get locked up out there in the

county. And they don't just slap you on the wrist. They put in for example—they make example out of you with drugs in the county. I don't want no part of it.

Interviewer: How about in the city? Do you ever feel that way in the city, needing to rush?

Tanya: Not as bad. I mean, I rush anyway, but not near as much […] Because I'm outside. I'm just paranoid. I'm a paranoid person. I don't want to get the cops called on me at all.

Having privacy in public and semi-public settings—which meant using alone for many participants—was also perceived to increase the risk of overdose, robbery, incarceration, and physical or sexual violence. For some participants, there was cognitive dissonance between their desire for privacy when using drugs in public and semi-public spaces and their perceptions of overdose risk factors. Inconsistencies in public and semi-public drug use behaviors and overdose risk perception are well illustrated in Mark's experience. When asked what he perceives to be a safe place to use drugs, Mark responded:

Alone. No people […] Anywhere where no human beings can see what I'm doing, that that can interfere what I'm doing.

However, he went on to describe how using drugs alone in a hidden space increases overdose risk:

So if I'm in an enclosed area, I think the [overdose] risk increases by 70%. I don't know why I said 70%, but it seems like a fair number [...] Because there's nobody to see me overdose because when you over—overdose means you die, you die basically [...] If I'm in a place that's vacant, there's nobody see that and it's tick tock.

Here, Mark's portrayal of safety is at odds with his earlier description of a 'safe' place to use drugs. His contradictory perception of safety underscores the duality of risks and privacy in public and semi-public drug use spaces.

**Subtheme 1.3: Buffering effect of social dynamics in public and semi-public spaces.** Social interactions were found to play a unique role in the tension between privacy and safety in public and semi-public drug use spaces. Gary intimated that using with other PWUD creates an emotionally safe environment where he can be his full self:

I mean, there's people who also get high sometimes [in abandoned buildings and on public transit]. I like the fact that we're all screwing up, but you don't have to ever worry about just trying to think of a lie on the spot, like talking to your family, like what you were doing. If you want to talk about drugs, you can be completely honest with anybody in there if you want to talk to them because nobody's going to judge you for getting high because we're all getting high.

For Gary, there's a sense of comfort and acceptance when among other PWUD. Though PWUD may ask to use his drugs and interrupt his 'peace' when high, they also serve as a source of informational support:

[...] when it comes to using and using around other people or where other people are and in abandons and stuff, it's the only place where I can actually talk about this stuff, if it's bothering me or anything. And other people usually will chime in and say, "Yeah, I had a situation like that. And I did dah dah dah dah dah." And that's what I like about it. You know what I mean? That's what helps.

Participants in intimate relationships often referenced using drugs with their intimate partner as strengthening their sense of safety. For Tanya, considerations of 'safety' extended beyond overdose risk to include the risk of robbery and sexual violence in public and semi-public spaces. She explained the dangers of using outside, particularly when in an opioid-induced drowsy state (i.e., nodding off):

> I've had people rob me, take everything that I had. Luckily, I haven't been molested or raped or anything like that. Because I've always had him [boyfriend] around. But I've known people that it's happened to. They nod out and wake up naked all their shit called […] But it's just so dangerous. You never know because you're not out on some of this shit. You could be out, out for a while and nothing's going to wake your ass up.

Importantly, her boyfriend's presence seems to temper her fear of being physically vulnerable when incapacitated. Towards the end of the excerpt, there is a notable shift in Tanya's thinking from her own potential risk of gender-based violence to the dangerous and unpredictable nature of drug use and its effects more generally. A strong fear of gender-based violence, particularly sexual violence, was common among female participants and was influential in how they perceived safety when sleeping and using drugs in public and semi-public places. Like Tanya, Edith expressed concern about the pervasive risk of gender-based violence when on the streets, and it influenced her perception of the risks associated with privacy and seclusion in unsheltered sleeping places. Edith reported typically sleeping in an enclosure near a public library with a group of women, in whom she seeks community and protection from violence. In addition to having a physical barrier separating her from other people in public spaces, being "near the women" figured prominently in her perception of a safe unsheltered sleeping place:

> You have to be near people. You don't want to be somewhere by yourself […] Because a man can come out and rape you and nobody would know. Or do something to you. At least if you're closer to the women, if you scream, they'll hear you.

### Theme 2: Experience accessing private spaces

In addition to re-appropriating public and semi-public spaces, some participants described experiences accessing private spaces for shelter and drug use. Commonly reported private spaces include cars and occupied houses belonging to other people. To access these private spaces, participants reported engaging in an informal economy of sharing items that most often included drugs and, in some cases, money or other things. All participants who had access to private spaces preferred them over public and semi-public spaces due to the increased sense of privacy, safety, and comfort. However, accessing these spaces was not without its risks, including becoming involved in other people's illicit activities. Also important, participants' level of trust towards other people occupying private spaces influenced their sense of safety in those settings.

**Subtheme 2.1: Accessing private spaces through an informal economy of sharing.** For all participants who had access to private spaces, access was inconsistent and contributed to their housing instability. Factors that influenced their access included people's schedules, community surveillance, and the amount of drugs or money they had. For example, throughout her interview, Tanya described accessing her friend's house to not only use drugs but also sleep and meet other basic needs, such as bathing. She went on to explain circumstances when she was unable to access the space:

> Whether he can have me over or not, if he's going to be busy the next day or his wife's going to be home or whatnot. If his wife's going to be home, sometimes he's like, "Well, nobody can be here just in case." She doesn't know I'll come over.

Accessing her friend's house is made complicated by the secretive nature of the informal economy of sharing. She must navigate informal arrangements to access the private space. Frank similarly highlighted the precarious nature of accessing his friend's car for shelter. His account draws attention to the often-fraught dynamics between neighborhood

residents and PEH in the neighborhood, particularly those who are unsheltered and therefore visibly inhabiting non-private spaces:

> Probably three months ago or something, slept in a car that my friend—a friend of mine's old car that he had parked around the corner from his house [...] the car winding up getting towed because the neighbors complained that they'd seen people going in and out of it.

Frank went on to discuss how his experience accessing private spaces, particularly private homes, was also contingent upon his ability to exchange drugs or pay rent:

> It was rough because it [house] wasn't mine, so of course you have to follow their rules, and sometimes they'll get upset because you don't have enough drugs for them or when you didn't have enough money to pay the rent. So it's kind of rough. So they either put you out or sleep for one night, and then you have to find someone else to go the next night.

For Frank, the experience of seeking private shelter is layered, as the use of 'rough' takes on a dual meaning. In addition to managing interpersonal dynamics when exchanging drugs or money for access to a private space, Frank notes that he must also compromise his autonomy. Consistent across interviews, accessing shelter off the streets involved a trade-off between an individual's agency or independence and a private place of refuge.

Another prominent aspect of participants' experiences accessing private spaces through the informal economy of sharing, touched upon in Frank's account, was an expectation that they give something to their friend or acquaintance in exchange for using the private space. As Robert stated:

You always give it [drugs] to the house. You always give something to the house. That's kind of like payment for that person allowing you to sit there and use.

Suggested by Robert's use of the word, 'payment,' it may be that Robert views the process of giving drugs to access a person's house as transactional. Though in the above passage he is unequivocal about 'always' paying, he later goes on to suggest that his level of closeness to someone determines the expectation of exchange—or lack thereof. When asked about situations in which he has been granted access to someone's home without payment, he replied:

> Certain ones, like if they're a really good friend or say, for instance, a girl you're dating or something like that. You will bring them something, but if you only had enough for yourself, you're not going to sit there and—I'll go upstairs and go to the bathroom and do it, but I'm not going to sit there in a person's face that uses as well as me and do drugs in there. Even if I only have a little bit, I'll still offer them something.

Robert's perspective on payment shifts throughout the passage. He begins by discussing circumstances where he does not exchange drugs and uses alone as a result, but concludes his stated desire to offer even a limited amount of his drugs. This seems to suggest a sense of obligation that may stem from his concern about straining relationships. Indeed, when asked why he alternates between sleeping at his friend's house and unsheltered situations, Robert expressed concern about being an imposition:

> No, because you should think about that [overstaying your welcome]. If a person is letting you stay there a couple of nights free of charge, you don't want to overdo it because I mean you absolutely, positively need that person. You don't want them to be upset with you or say, "Well, the last time you were here, such and such and such happened." So you don't overstay your welcome. You don't burn bridges.

Evident in a few participants' experiences was the mutually beneficial nature of the informal economy of sharing. There was a sense of mutuality, for example, in Tanya's relationship with her friend who gives her and her boyfriend access to his house:

> We'll sleep on his floor, which sucks. But he kind of looks out for us a little bit and everything. And he needs the help too right now. He's retired. He don't make no money and everything like that. So it's a good situation for both of us. We help him out and he helps us out [...] he'll let us take a shower at his house and he'll feed us sometimes. And I help him support his habit more or less.

**Subtheme 2.2: Risks and benefits of private spaces.** Common throughout participants' accounts was the claim that private spaces created safer, more comfortable environments for drug use. These spaces enabled participants to engage in harm reduction and sanitary drug use practices, as well as be discerning about who was present in the space. Several participants' accounts also pointed to the social dimension of overdose prevention, noting how using drugs in friends' houses and cars reduced their non-fatal overdose risk. We will first focus on Tanya's account of using drugs at her friends' houses, as her account captures many aspects of other participants' experiences using in private spaces. By contrasting her experiences using outside, Tanya highlights the varied ways in which using inside her friends' home offers physical safety:

> I could take my time and not try to rush it, and then I don't hurt myself as much [...] it's warm [inside the house], first off. And like I said, when it's cold, my veins don't work at all, and it takes so long to be able to get on. And I just keep hurting myself because I can't find the vein. […] It's a lot cleaner. I can get actual faucet water instead of gutter water or whatever I have to scrape up. So that's a plus. [...] If I nod off, me and him [boyfriend]. If we nod out or whatever, at least we're in a safe area. We're not outside where somebody could stumble across us and rob us or something could happen.

Many participants also communicated a similar desire for warmth in both drug use and sleeping places, suggesting its saliency for the experiences of unsheltered PEH who use drugs.

Pointing to the social dimension of fatal overdose prevention within private settings, Tanya and several other participants emphasized the important role friends played in intervening with naloxone or calling for emergency medical services should they experience an overdose. Tanya described a hypothetical scenario during which she overdosed while at her friend's house:

> He [friend] wouldn't let anything happen. If something happened, he'd call 911, and I'd get help regardless. He doesn't know what to do. My friend doesn't. He don't know the first thing about Narcan and somebody or anything like that, but say something happened with my boyfriend, he had to step outside or something, and I overdosed. My friend would not just let me die.

A high level of trust in her friend seems to be an essential part of why Tanya feels safer using at her friends' house. For many participants who reported using in private spaces, a sense of trust was critical in shaping their sense of safety in the private space. In Robert's case, for example, there was a distinction to be made between people who increased his sense of safety and people who threatened his sense of safety based on how much he trusted them. His strong concern about becoming involuntarily involved in other people's illegal activities shaped his decision making about where to use and who to use with in private spaces. Here Robert explains the severe consequences of using around people he does not know well:

Because I don't feel comfortable using around people that I don't know [...] You don't know what that person has going on in the streets. So I could be hanging out with a person and not knowing that the whole time this person is running from somebody that they owe money to or drugs to, and that person finds out where they are. Now the door's getting kicked in and everybody's getting shot or hurt or something just for something that somebody did and they didn't let you know what their involvements on the streets.

**Theme 3: Desired drug use spaces: Overdose prevention site**

When prompted about their ideal place to use drugs safely and comfortably, participants articulated a desire for access to supplies that enable hygienic drug use and resources that reduce their risk of fatal overdose. Some participants referenced overdose prevention sites (OPSs) in Canada as a potential model for their ideal drug use space, and many spoke of resources available at OPSs without explicitly using the term, OPS. Desired resources in a place for drug use included private rooms, peer specialists, medical personnel, sterile syringes, cookers, and cotton. In addition to resources that enable safe consumption of drugs, some participants wanted resources that addressed their other needs, including housing and substance use treatment. When asked about his ideal place to use drugs, Gary explained:

They do it in, I think, Vancouver or something. You walk into a place like this [study clinic] or whatever. And they have a nurses station. They sign you in. But when you go into the back, there's a big bin, a box in the middle, and they're full of syringes. They're new ones. And then another box may have cookers, another box has the ties and cotton. Anything you need […] And there's little rooms serving numbers like one, two, three. Those rooms have glass and a lot of lighting. And they even have a thing where they provide a nurse's help to get you on if you can't do it and it's free. And what they do is they do that and so instead of somebody messing around and screwing their body up for like two hours or whatnot digging around, they can go to a place like that.

Frank similarly spoke about the critical role of nurses in a supervised place for safe drug consumption, particularly "to help you out if you do overdose." When asked where he would like the place to be located, Frank elaborated:

They should have one on the east side and on the west side, somewhere near Johns Hopkins and another one near a hospital on the west side […] I think it'd be better because if something was to happen, you have nurses, and it's just a better area for it.

Interviewer: Yeah. Would you want it to be close to where you're sleeping right now?

Frank: No […] It's just because you don't want a lot of people coming to where you're trying to rest at or whatever […] Because then they might just wind up coming and staying where you're at.

While many participants wanted the supervised site to help facilitate safer drug consumption, Eric held a contrasting view on the purpose of such a place. He felt that they should help people transition away from drug use:

But the thing is, the whole goal is to build it where it's going to be a transition of not using, you know what I'm saying? Stasis. You might come in and they got it real bad. And you have the clinical doctors and things of that nature there. And the goal is to transition. Levels of getting them not to use.

It is also important to note that although there was strong support for a supervised site for safe drug consumption among several participants, support was not universal. As noted in his response below, Robert was skeptical of the possibility an OPS would work out due to violence caused by other PWUD in Baltimore City:

It would be too dangerous. Baltimore City, it probably would never work. It would never work […] Because this is Baltimore City. Somebody will mess it up […] Not a cop. Because if it's a legal place where you—a cop wouldn't have any jurisdiction. But somebody would mess it up. Somebody would take somebody's stuff or something, or somebody would wind up getting shot or something over something stupid. They would mess it up.

## Discussion

This study explored how unsheltered PEH interpret features of their micro drug use environments and manage their drug use practices and overdose risk across public, semi-public, and private settings. To our knowledge, this is the first qualitative study designed to examine drug use experiences specific to unsheltered PEH. Participant experiences revealed that privacy and overdose risk were inherently in competition across all settings—public, semi-public, and private—and that participants' prevailing desires for drug use spaces hidden from the police, other people who use drugs, and the general public often superseded considerations of drug use safety. Social interactions and trust towards people present in public, semi-public, and private drug use environments notably shaped participants' perceived vulnerability to overdose and violence in such spaces. When describing an imagined space to use drugs safely, many participants described a building—akin to an overdose prevention site—that was proximal to a hospital, offered supervised areas for safe drug consumption, had embedded medical staff, and provided sterile drug use equipment.

Participants expressed an unequivocal desire for privacy in their public and semi-public drug use spaces. However, participants' search for privacy in public and semi-public spaces was often at the expense of their engagement in harm reduction practices. Our findings mirror results from existing studies on PWUD's motivations for using alone, which similarly included drug use-related stigma and not wanting to share drugs [57–59]. Additionally, study findings align with prior epidemiologic studies demonstrating that injecting in public and semi-public spaces is associated with a significantly higher odds of non-fatal overdose [25,37,39]. Our study participants' experiences point to potential underlying mechanisms for this association—that fear of police interactions and arrest may result in participants using alone and rushing their drug use [60–62]. Similar consequences of policing on high-risk drug use behaviors have been found among people who inject drugs (PWID) in Canada and the United Kingdom [32,41]. Important to note is that all participants in our study were constrained to using drugs in public and semi-public spaces due to their housing context; thus, the need to manage competing priorities was an ever-present challenge and may be felt more acutely among unsheltered PEH who use drugs than those who are experiencing sheltered homelessness or are housed.

A prominent aspect of participants' experience with public and semi-public drug use that is less explored in the literature is the dual role of police in creating harm and safety. In the context of their drug use, participants expressed fear of police interactions. In the context of their unsheltered homelessness, however, many had a positive perception of police and some preferred sheltering with police nearby to serve as a deterrent from crime and people who may threaten their physical safety. Consistent across interviews was an understanding that third parties, particularly community residents and Johns Hopkins, may cause subsequent law enforcement policing in public and semi-public spaces. This approach to policing, which involves "call-driven reactive policing" initiated by a third party (e.g., community members, businesses, city agencies, public officials) was recently coined by Herring as complaint-oriented policing and found to similarly characterize dynamics between police and PEH in San Francisco [63]. Given this study's findings and similar policies criminalizing visible homelessness and survival behaviors PEH engage in (e.g., panhandling) across North American and European countries [64], additional research is critically needed to examine how anti-homelessness policies shape policing practices towards unsheltered PEH, and in turn, create high-risk drug use environments is critically needed. Particular attention should be paid to the role of community and local businesses in driving policing efforts and creating hostile environments to visible homelessness and drug use.

Private, occupied houses are common settings where fatal overdoses occur [65,66]. However, in many ways, the physical and social features of private drug use spaces (e.g., car, occupied house) enhanced participants' ability to use drugs safely and minimized the risk of sequelae associated with rushed and unhygienic drug use. In particular, some participants described how using drugs in a private space with a trusted person fostered a safer drug use environment by minimizing the risk of a fatal overdose and of unwanted interruptions by police or people who may steal from them. In a qualitative study of young PWID, Winiker et al. similarly found that to mitigate risks associated with drug use, participants used with trusted friends who they know would look out for their safety [62]. In contrast to the present study, Winiker et al. did not distinguish between the social dynamics in public or semi-public drug use spaces and private drug use spaces. Additional investigation is needed to understand how unsheltered PEH leverage their limited social networks to access and safely use drugs in private drug use spaces.

As overdose deaths continue to escalate in the US, some cities have considered implementing OPSs to provide PWUD with private, hygienic spaces to use drugs safely with staff on site to respond to an overdose and connect clients with social, medical, and harm reduction services. A robust literature base has found that OPSs in Canada and Australia are associated with reduced overdose mortality [67], opioid-related overdoses [68], and sharing of syringes or equipment [69–71], as well as increased access to substance use treatment and ability to engage in hygienic drug use practices [72–75]. Moreover, OPSs have been shown to be associated with no increases or reductions in crime and public injections [75–78]. Although an OPS has not yet been implemented in Baltimore City, many study participants spoke about the benefits of having access to a building for supervised, hygienic drug use and described such a building as an ideal place to use drugs safely. Our study findings highlight additional considerations and challenges for the implementation of OPSs in similar settings, including clients' varying perspectives on harm reduction, abstinence, and concerns about community violence.

Study findings point to the need for peer-led and peer-driven harm reduction and low barrier housing programs that center the lived experiences of unsheltered PEH who use drugs. While critical to promoting health equity and creating effective programs that better reflect community needs, involving peers in the development and implementation of harm reduction programs also helps build trust and reduce internalized drug use-related stigma [79–83]. Additionally, drawing on the Housing First approach, low barrier permanent supportive housing that offers long-term housing without preconditions of abstinence or engagement in drug treatment is a promising approach to supporting the well-being of unsheltered PEH, including improving their overall health [84], decreasing hospitalizations [85,86], and reducing mortality [86]. On the contrary, police-involved post-overdose outreach and diversion programs leveraging coercive tactics to engage PWUD in harm reduction and treatment services have been implemented in the US [87–90]—despite some evidence linking policing with reduced access to syringe service and treatment programs [88,91–93]. Taken together, coercive tactics and existing mistrust towards law enforcement, as highlighted in our study, may exacerbate mistrust and further contribute to high-risk drug use behaviors.

Study findings should be interpreted in light of a few limitations. Participant experiences are situated within the context of Baltimore City, a hyper-segregated, post-industrial city with high rates of drug use and overdose. Participants were also recruited from a prospective cohort study of people with a history of injection drug use, whose engagement with the study may be a marker of increased engagement in, or referral to, health services and resources. Therefore, the transferability of findings may be limited to individuals who use drugs and live in similar urban contexts in the US, and those who have greater engagement with health and social services. However, findings from this study also aligns with evidence elsewhere, which suggests that increased police activity similarly promotes high risk drug use behaviors and hinders engagement in harm reduction strategies across diverse settings internationally [41,42,91,94]. Drug use and homelessness are highly sensitive topics; some participants may have been hesitant to speak openly about their experiences due to stigma and fear of legal repercussions. Our study also did not formally conduct member checking but completed respondent validation during the data collection process. Therefore, some dimensions of drug use and unsheltered homelessness may

have been missed during data collection and data analysis, affecting the credibility of our findings. However, the ALIVE study has been ongoing for over 30 years, and ALIVE staff have established rapport and mutual trust with participants—thereby fostering a destigmatizing and comfortable environment for individuals to participate in research studies. During interviews, participants demonstrated comfort in speaking at length and sharing extensive detail about their experiences with drug use and homelessness. While AQT and BD do not have personal lived experience with homelessness and drug use, they are informed "outsiders" with backgrounds in substance use and public health and experience working with socially marginalized populations, including unsheltered populations. Being an informed "outsider" may have also encouraged participants to describe their experiences in detail during their interviews, with the assumption that AQT is not knowledgeable on the lived realities of unsheltered PEH who use drugs. Further, as the first to explore micro drug use environments specific to unsheltered PEH in the US, this study provides timely insight into their dual experiences of visible homelessness and drug use and their unique challenges navigating drug use practices and overdose risk.

## Conclusion

Findings highlight that unsheltered PEH who use drugs must balance their desire for privacy with overdose risk across drug use settings, thus compromising the efficacy of individual-level interventions encouraging people to never use alone and to use slowly [95]. Participant experiences can help inform multi-level interventions that promote harm reduction and reduce overdose risk, including the creation of OPSs in the US [95].

## Acknowledgments

The authors would like to acknowledge ALIVE study staff who assisted with recruitment and thank the research participants who shared their experiences and trusted us with their stories.

## Author contributions

**Conceptualization:** Ashley Q Truong, Haneefa T Saleem, Jessie Chien, Sabriya L Linton.

**Data curation:** Ashley Q Truong.

**Formal analysis:** Ashley Q Truong, Bridget Duffy.

**Funding acquisition:** Ashley Q Truong, Gregory D Kirk, Shruti H Mehta, Becky L Genberg.

**Investigation:** Ashley Q Truong, Bridget Duffy, Jessie Chien, Sabriya L Linton.

**Methodology:** Ashley Q Truong, Bridget Duffy, Haneefa T Saleem.

**Resources:** Ashley Q Truong.

**Supervision:** Haneefa T Saleem, Becky L Genberg, Sabriya L Linton.

**Visualization:** Ashley Q Truong.

**Writing – original draft:** Ashley Q Truong, Bridget Duffy, Haneefa T Saleem, Sabriya L Linton.

**Writing – review & editing:** Ashley Q Truong, Bridget Duffy, Haneefa T Saleem, Jessie Chien, Gregory D Kirk, Shruti H Mehta, Becky L Genberg, Sabriya L Linton.

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
