## [Decision Letter · Decision Letter 0]

24 Jun 2025

Dear Dr. Truong,

Thank you for submitting your manuscript to PLOS ONE. After careful consideration, we feel that it has merit but does not fully meet PLOS ONE’s publication criteria as it currently stands. Therefore, we invite you to submit a revised version of the manuscript that addresses the points raised during the review process.

**ACADEMIC EDITOR:**
**Please address the reviewer comments and resubmit for further consideration for publication**

We look forward to receiving your revised manuscript.

Kind regards,

Souparno Mitra, M.D.

Academic Editor

PLOS ONE

 [This study was funded by the National Institute on Drug Abuse [U01DA036297, PIs: Mehta/Kirk; R01DA053136, PI: Genberg], the Johns Hopkins Bloomberg School of Public Health Center for Qualitative Studies in Health and Medicine’s Dissertation Enhancement Award, and The Paul V. Lemkau Scholarship Fund. AQT was supported by the National Institute on Drug Abuse [T32DA007292-31A1; PIs: Johnson/Maher].]. 

Additional Editor Comments (if provided):

Reviewers' comments:

Reviewer's Responses to Questions

**Comments to the Author**

1. Is the manuscript technically sound, and do the data support the conclusions?

Reviewer #1: Yes

Reviewer #2: Yes

2. Has the statistical analysis been performed appropriately and rigorously?

Reviewer #1: I Don't Know

Reviewer #2: Yes

3. Have the authors made all data underlying the findings in their manuscript fully available?

Reviewer #1: Yes

Reviewer #2: Yes

4. Is the manuscript presented in an intelligible fashion and written in standard English?

Reviewer #1: Yes

Reviewer #2: Yes

Reviewer #1: Well written article- I have attached the PDF of suggested edits.

I have also included other limitations that need to be added -

1) Small sample size

2) Participants were drawn from the AIDS Linked to Intravenous Experience (ALIVE) Study, an ongoing cohort study. This could result in a selection bias. as being involved in long-term research may differ from those who are not engaged with such services or institutions.

3) The study relies on self report. Probable biases include recall bias, and underreporting of risky behaviors due to stigma or legal fears.

4) The study relies only on qualitative interview data, without integrating observational data or input from service providers, which could have strengthened the findings.

Reviewer #2: Line 26: Due to their economic and social marginalization. Does social margination includes economic?

Line 31: Please define public, semi-public, and private places

Line 32: AIDS Linked to Intravenous Experience Study “(ALIVE)”

Line 81: people who inject drugs (PWID)

Line 72: and temporarily/transitionally housed. Refer to line 114/5, which divides into unsheltered, sheltered, and transitional housed.

Line 123: 3 people were excluded due to mental health- tell us more because substance abuse is also a mental health disorder

Line 127: speedball (cocaine + opioid?)

Line 130: Why were sheltered people excluded, because this study focused on unsheltered with PEH?

Line 423: Why is part of the conversation in quotes?

**Do you want your identity to be public for this peer review?** For information about this choice, including consent withdrawal, please see our Privacy Policy

Reviewer #1: **Yes: ** Arun Prasad

Reviewer #2: **Yes: ** Anoop Narahari

---

## [Author Response · Author response to Decision Letter 1]

28 Aug 2025

Response to Reviewers for “Navigating public, semi-public, and private drug use environments: A qualitative examination of unsheltered homelessness and drug use in Baltimore City” [PONE-D-25-23824]

Dear Dr. Mitra,

Thank you for the opportunity to revise this manuscript. We thank the reviewers for their thoughtful comments and have responded to each comment, as outlined below.

COMMENTS FOR THE AUTHOR:

From the Editor –

RESPONSE: The manuscript has been revised to meet PLOS ONE’s style requirements. Appropriate level headings have been applied to the manuscript sections. The naming convention for Figure 1 in line 170 on page 9 of the unmarked revised manuscript has been revised to “Fig 1”. The table 1 formatting has been updated to meet the style requirements. In-text citations have been updated to cite references in brackets. Funding information has been removed from the acknowledgements, and the following statement has been added in the Acknowledgements section in lines 671-672 on page 30 of the unmarked revised manuscript: “The authors would like to acknowledge ALIVE study staff who assisted with recruitment and thank the research participants who shared their experiences and trusted us with their stories.”

[This study was funded by the National Institute on Drug Abuse [U01DA036297, PIs: Mehta/Kirk; R01DA053136, PI: Genberg], the Johns Hopkins Bloomberg School of Public Health Center for Qualitative Studies in Health and Medicine’s Dissertation Enhancement Award, and The Paul V. Lemkau Scholarship Fund. AQT was supported by the National Institute on Drug Abuse [T32DA007292-31A1; PIs: Johnson/Maher].].

RESPONSE: The funders had no role in the study design, data collection and analysis, decision to publish, or preparation of the manuscript. This statement has been added to the final paragraph of our cover letter.

RESPONSE: There are ethical restrictions on sharing de-identified data from this study because the data contains potentially identifying and sensitive patient information. These restrictions are imposed by the Johns Hopkins Institutional Review Board. De-identified data requests may be sent to the ALIVE study PIs, Shruti Mehta (smehta@jhu.edu) and Gregory Kirk (gdk@jhu.edu), or the Johns Hopkins Bloomberg School of Public Health IRB office (BSPH.irboffice@jhu.edu). The Data Availability statement has been updated accordingly.

RESPONSE: A separate caption has been added for Figure 1 in line 177 on page 9 of the unmarked revised manuscript.

RESPONSE: We have reviewed the reference list and confirm it is complete and correct.

From the Reviewers –

Reviewer 1:

1. Well written article- I have attached the PDF of suggested edits.

I have also included other limitations that need to be added –

1) Small sample size

2) Participants were drawn from the AIDS Linked to Intravenous Experience (ALIVE) Study, an ongoing cohort study. This could result in a selection bias. as being involved in long-term research may differ from those who are not engaged with such services or institutions.

3) The study relies on self report. Probable biases include recall bias, and underreporting of risky behaviors due to stigma or legal fears.

4) The study relies only on qualitative interview data, without integrating observational data or input from service providers, which could have strengthened the findings.

RESPONSE: Thank you for including suggested edits in the PDF. I have transferred the suggested edits and line numbers in this document, as well as included our response to the suggested edits.

2. Line 33: Sample size seems to be too small at 9.

RESPONSE: Thank you for your comment. This study was conceptualized and carried out using interpretative phenomenological analysis (IPA). Lines 131-134 on page 7 of the unmarked revised manuscript have been edited to convey the justification for small sample sizes in IPA studies: “According to Smith et al., sample sizes in studies using IPA are necessarily small due to the approach’s focus on understanding the particularities of participants’ daily lived experiences, and to enable deep analysis of each interview [52]. As such, larger sample sizes are typically discouraged in IPA studies, and recommended sample sizes range from 4-10 [53, 54].”

Due to this study’s use of IPA methodology, we believe that the sample size of 9 is appropriate for the qualitative method used in this study and sufficient to capture the phenomenon under study while also allowing for a detailed analysis of the particularities in each participant’s lived experiences of homelessness and drug use.

3. Lines 120-124: The small sample size significantly reduces the power of the study. This makes sense for an IPA study, however would have loved to see a higher sample size of around 30-35 which could have definitely have been done given the large size of the population being studies here.

RESPONSE: Given the parent study we sampled participants from, it was likely possible to sample a larger number of individuals who were experiencing unsheltered homelessness and using drugs. However, we employed sampling procedures that were in line with IPA because our study was conceptualized and carried out using this method [1, 2]. As specified in response to your second comment, small sample sizes are necessary in IPA studies to enable a detailed analysis of the nuances in participants’ daily lived experiences around the phenomenon under study. Larger sample sizes are typically discouraged in IPA studies, as they do not facilitate detailed analyses of participants’ experiences.

4. Line 136: An objective drug testing would have been a more reliable and objective method.

RESPONSE: Thank you for your comment. While we recognize that drug testing enables the detection of drugs in an individual’s system, we chose to rely on self-reported drug use for a number of reasons. First, as noted in line 156 on page 8 of the unmarked revised manuscript, we asked about their drug use in the past 30 days during the eligibility screening. During the interview, participants spoke extensively about their drug use experiences and were probed to speak in further detail about those experiences. Thus, our concern about the participant misremembering or forgetting their drug use was minimized. Second, we recruited participants from the AIDS Linked to IntraVenous Experience Study, which has been ongoing for over 30 years. ALIVE staff have established strong rapport and mutual trust with participants over this period. Recruiting participants through the ALIVE study and with support from ALIVE staff created a trusting environment for participants to more openly and comfortably discuss sensitive topics, such as their drug use. Third, constructivism is the epistemology that underlies qualitative research. From the constructivist lens, knowledge is co-created by the researcher and the participant, and what a participant expresses to the researcher is perceived as the participant’s reality or “truth.” In contrast, objectivism is the epistemology that underlies quantitative research. Objectivism prioritizes being “objective” and reducing bias. Therefore, due to this study’s use of qualitative methods, we believed asking participants to self-report their drug use was appropriate and sufficient to determine their eligibility in the study. Fifth, the use of self-reported drug use is in line with what previous qualitative studies with people who use drugs have used to determine eligibility, including qualitative studies that have recruited from the ALIVE study [3-6].

5. Line 615: Limitations to be included-

1) Small sample size

2) Participants were drawn from the AIDS Linked to Intravenous Experience (ALIVE) Study, an ongoing cohort study. This could result in a selection bias. as being involved in long-term research may differ from those who are not engaged with such services or institutions.

3) The study relies on self report. Probable biases include recall bias, and underreporting of risky behaviors due to stigma or legal fears.

4) The study relies only on qualitative interview data, without integrating observational data or input from service providers, which could have strengthened the findings.

RESPONSE: Thank you for your suggestions on additional limitations to be included. Below are our responses to each of the limitations you identified.

Limitation 1

As indicated in our responses to your second and third comments related to the study’s sample size, we have addressed your concerns by including additional rationale for the sample size used and why the sample size is appropriate for an IPA study in lines 131-134 of the unmarked revised manuscript. IPA studies recommend sample sizes of 4-10 [1, 2].

Limitations 2 and 3

We have incorporated your second and third suggested limitations in lines 637-660 on pages 28-29 of the unmarked revised manuscript. In the manuscript, we discussed your proposed second and third limitations using the trustworthiness criteria for qualitative studies, which is often used to measure rigor in qualitative studies [7, 8]. Traditional scientific criteria applied in quantitative studies (i.e., internal validity, external validity or generalizability, reliability, objectivity) is not well-suited for qualitative studies due to differences in philosophical underpinnings of what knowledge is and how it is obtained in qualitative and quantitative research [8]. In other words, there are differences in the epistemology of qualitative research (i.e., constructivism) and epistemology of quantitative research (i.e., objectivism) that make trustworthiness criteria a more suitable indication of rigor in qualitative research. According to the trustworthiness criteria, credibility is analogous to internal validity, transferability is analogous to external validity, dependability is analogous to reliability, and confirmability is analogous to objectivity [7, 8]. We address your second concern about selection bias as a limitation of the transferability of our findings in lines 637-641 of the unmarked revised manuscript: “Participants were also recruited from a prospective cohort study of people with a history of injection drug use, whose engagement with the study may be a marker of increased engagement in, or referral to, health services and resources. Therefore, the transferability of findings may be limited to individuals who use drugs and live in similar urban contexts in the US, and those who have greater engagement with health and social services.” We address your third concern about recall bias and underreporting as a limitation of the credibility of our findings in lines 644-660 of the unmarked revised manuscript: “Drug use and homelessness are highly sensitive topics; some participants may have been hesitant to speak openly about their experiences due to stigma and fear of legal repercussions. Our study also did not formally conduct member checking but completed respondent validation during the data collection process. Therefore, some dimensions of drug use and unsheltered homelessness may have been missed during data collection and data analysis, affecting the credibility of our findings. However, the ALIVE study has been ongoing for over 30 years, and ALIVE staff have established rapport and mutual trust with participants—thereby fostering a destigmatizing and comfortable environment for individuals to participate in research studies. During interviews, participants demonstrated comfort in speaking at length and sharing extensive detail about their experiences with drug use and homelessness. While [authors blinded for review] do not have personal lived experience with homelessness and drug use, they are informed “outsiders” with backgrounds in substance use and public health and experience working with socially marginalized populations, including unsheltered populations. Being an informed “outsider” may have also encouraged participants to describe their experiences in detail during their interviews, with the assumption that [author blinded for review] is not knowledgeable on the lived realities of unsheltered PEH who use drugs.”

Limitation 4

Although this qualitative study filled a gap in research on the drug use experiences of people who are unsheltered by gathering perspectives of people with lived experience, it was beyond the scope of this study to understand perspectives of providers from behavioral health and housing sectors. Insight from these groups could have identified potential structural and programmatic aspects of drug use and overdose experiences that may not have otherwise been understood by exploring the perspectives of people with lived experience. Likewise, while this study did not have a multi-method design, future quantitative analysis can generate empirical evidence on relationships of risk environment features (e.g., policing) and engagement in harm reduction practices with overdose.

Reviewer 2:

1. Line 26: Due to th

---

## [Decision Letter · Decision Letter 1]

20 Nov 2025

Navigating public, semi-public, and private drug use environments: A qualitative examination of unsheltered homelessness and drug use in Baltimore City

PONE-D-25-23824R1

Dear Dr. Truong,

We’re pleased to inform you that your manuscript has been judged scientifically suitable for publication and will be formally accepted for publication once it meets all outstanding technical requirements.

Kind regards,

Souparno Mitra, M.D.

Academic Editor

PLOS ONE

Additional Editor Comments (optional):

Reviewers' comments:

Reviewer's Responses to Questions

**Comments to the Author**

Reviewer #1: All comments have been addressed

Reviewer #2: All comments have been addressed

2. Is the manuscript technically sound, and do the data support the conclusions?

Reviewer #1: Partly

Reviewer #2: Yes

3. Has the statistical analysis been performed appropriately and rigorously?

Reviewer #1: I Don't Know

Reviewer #2: Yes

4. Have the authors made all data underlying the findings in their manuscript fully available?

Reviewer #1: Yes

Reviewer #2: Yes

5. Is the manuscript presented in an intelligible fashion and written in standard English?

Reviewer #1: Yes

Reviewer #2: Yes

Reviewer #1: Thank you for the edits on "Navigating public, semi-public, and private drug use environments: A qualitative examination of unsheltered homelessness and drug use in Baltimore City."

Reviewer #2: All the eight reviewer 2 comments were addressed and necessary edits were made. I prefer conclusion not have references.

**Do you want your identity to be public for this peer review?** For information about this choice, including consent withdrawal, please see our Privacy Policy

Reviewer #1: No

Reviewer #2: **Yes: ** Anoop Narahari

---

## [Editor Report · Acceptance letter]

PONE-D-25-23824R1

PLOS ONE

Dear Dr. Truong,

I'm pleased to inform you that your manuscript has been deemed suitable for publication in PLOS ONE. Congratulations! Your manuscript is now being handed over to our production team.

Kind regards,

on behalf of

Dr. Souparno Mitra

Academic Editor

PLOS ONE